# Anticancer Activity and Molecular Mechanisms of an Ursodeoxycholic Acid Methyl Ester-Dihydroartemisinin Hybrid via a Triazole Linkage in Hepatocellular Carcinoma Cells

**DOI:** 10.3390/molecules28052358

**Published:** 2023-03-03

**Authors:** Ya-Fen Hsu, Fan-Lu Kung, Tzu-En Huang, Yi-Ning Deng, Jih-Hwa Guh, Paolo Marchetti, Elena Marchesi, Daniela Perrone, Maria Luisa Navacchia, Lih-Ching Hsu

**Affiliations:** 1School of Pharmacy, College of Medicine, National Taiwan University, Taipei 10050, Taiwan; 2Department of Chemical and Pharmaceutical Sciences, University of Ferrara, 44121 Ferrara, Italy; 3Department of Environmental and Prevention Sciences, University of Ferrara, 44121 Ferrara, Italy; 4Institute for Organic Synthesis and Photoreactivity (ISOF), National Research Council of Italy (CNR), 44129 Bologna, Italy

**Keywords:** hepatocellular carcinoma, bile acid–dihydroartemisinin hybrids, anticancer, oxidative stress, autophagy, apoptosis

## Abstract

Hepatocellular carcinoma is the third most common cause of cancer-related death according to the International Agency for Research on Cancer. Dihydroartemisinin (DHA), an antimalarial drug, has been reported to exhibit anticancer activity but with a short half-life. We synthesized a series of bile acid–dihydroartemisinin hybrids to improve its stability and anticancer activity and demonstrated that an ursodeoxycholic–DHA (UDC-DHA) hybrid was 10-fold more potent than DHA against HepG2 hepatocellular carcinoma cells. The objectives of this study were to evaluate the anticancer activity and investigate the molecular mechanisms of UDCMe-Z-DHA, a hybrid of ursodeoxycholic acid methyl ester and DHA via a triazole linkage. We found that UDCMe-Z-DHA was even more potent than UDC-DHA in HepG2 cells with IC_50_ of 1 μM. Time course experiments and stability in medium determined by cell viability assay as well as HPLC-MS/MS analysis revealed that UDCMe-Z-DHA was more stable than DHA, which in part accounted for the increased anticancer activity. Mechanistic studies revealed that UDCMe-Z-DHA caused G0/G1 arrest and induced reactive oxygen species (ROS), mitochondrial membrane potential loss and autophagy, which may in turn lead to apoptosis. Compared to DHA, UDCMe-Z-DHA displayed much lower cytotoxicity toward normal cells. Thus, UDCMe-Z-DHA may be a potential drug candidate for hepatocellular carcinoma.

## 1. Introduction

According to the International Agency for Research on Cancer, primary liver cancer is the sixth most common cancer in the world and the third leading cause of cancer death for both sexes in 2020 [1]. Hepatocellular carcinoma (HCC) is the most common form and accounts for 80–90% of primary liver cancer [2]. In the early stages, HCC can be treated with surgical procedures. However, many patients are diagnosed at advanced stages and rely on non-surgical procedures, such as chemotherapy and radiation therapy, and the survival rate remains low [3,4]. Therefore, there is an urgent need to develop more effective treatment options.

Artemisinin, a sesquiterpene lactone containing an endoperoxide bridge, was originally isolated from the Chinese herb *Qinghao* (*Artemisia annua*) as an effective antimalarial component [5]. Artemisinin and its derivatives (artemisinins) have become the standard therapy for malaria. Recent studies revealed that artemisinins also displayed antitumor, antidiabetic, antifungal, immunomodulatory, antiviral, anti-inflammatory and antibacterial activities. However, a short half-life, and poor water solubility and bioavailability limited further anticancer applications of artemisinins [6].

Dihydroartemisinin (DHA), the reduced lactol derivative of artemisinin, is more stable and much more potent than artemisinin [5]. The presence of the hemiacetal moiety in DHA improves the water solubility of the parental artemisinin and offers the chance for further chemical modifications to obtain artemisinin derivatives. DHA is also the main active metabolite of the artemisinins. DHA has been shown to exert antitumor activity in a variety of cancer cells, including breast [7], colon [8], lung [9] and liver cancer [10]. Studies have shown that DHA exerts anticancer activities through various molecular mechanisms, such as inhibiting proliferation and inducing apoptosis, DNA damage, reactive oxygen species (ROS) and autophagy, etc. Due to the labile chemical structure of DHA, approaches including the development of DHA hybrids have been employed to improve its stability and anticancer activity [6].

Bile acids (BAs) are synthesized from cholesterol in the liver, and secreted into the bile, released into the intestine, and then recycled back to the liver. The enterohepatic circulation is very efficient, therefore, bile acids can be exploited for the design of prodrugs to improve intestinal absorption and metabolic stability, and even sustain the release of active drugs. Moreover, BAs can also increase cell membrane fluidity or serve as carriers for targeting drugs to the liver [11]. It has been reported that BAs display antitumor activity toward cancer cell lines, but the relatively low activity with IC_50_ greater than 100 μM prevents their use in cancer therapy [11,12]. Due to the unique physical–chemical properties, biological activity and the presence of different available hydroxyl and carboxylic moieties suitable for the formation of covalent bonds, BAs have become interesting templates for molecular hybridization with drug or non-drug molecules [13]. For instance, BAs conjugated with drugs such as tamoxifen [14], cytarabine [15] and paclitaxel [16] as well as functional group or aromatic bioactive synthetic units [17] have been reported and evaluated for their anticancer activity. Moreover, conjugates of BAs with natural molecules such as nucleosides [18] and artemisinins [19,20] have received much attention as potential anticancer agents.

In a previous work, we synthesized a library of bile acid–dihydroartemisinin (BA-DHA) hybrids by conjugating a series of BAs, including ursodeoxycholic acid (UDCA), and DHA through both cleavable and non-cleavable linkers. A cleavable linker is designed to release the two agents under physiological or enzymatic conditions (prodrug strategy), whereas a non-cleavable linker, keeping its structure intact, leads to a hybrid drug. The condensation reaction was applied to obtain BA-DHA ester derivatives, whereas the click reaction (the copper-catalyzed azide-alkyne cycloaddition Cu-AAC) was used to obtain a non-cleavable linkage through the formation of the 1,2,3-triazole ring. The triazole moiety has been shown to improve pharmacological, pharmacokinetic and physiochemical profiles of bioactive compounds [21]. Most of the BA-DHA hybrids displayed significant improvement in anticancer activity and some hybrids were at least 10 times more potent than DHA against HL-60 leukemia cells and HCC cells, demonstrating the effectiveness of the conjugation [19,20]. UDC-DHA, the ester hybrid obtained by the conjugation of UDCA at the C-24 position with DHA through a condensation reaction (Figure 1), has been further evaluated in HCC cells. UDC-DHA is found to be 10–20 times more potent than DHA in HepG2 and Huh-7 HCC cells and may act as a prodrug to improve the stability as well as sustain the release of the active drug DHA [20].

UDCMe-Z-DHA is a hybrid obtained by the conjugation of ursodeoxycholic acid methyl ester (UDCMe) at the C-3 position with DHA via a stable triazole linkage (Figure 1). We reported previously that UDCMe-Z-DHA (hybrid **11**) was equally potent as UDC-DHA (hybrid **2**) against HepG2 cells [19]. The objectives of this study were to evaluate the anticancer activity and determine the molecular mechanisms of UDCMe-Z-DHA purified using an improved procedure in HCC cells. We showed that UDCMe-Z-DHA displayed potent anticancer activity in HCC cells. We also demonstrated that UDCMe-Z-DHA caused cell cycle arrest, and induced ROS as well as mitochondrial membrane potential (MMP) loss, which may activate autophagy and in turn lead to apoptosis.

## 2. Results

### 2.1. The Anticancer Effect of UDCMe-Z-DHA Is More Potent Than DHA in HCC Cells

The UDCMe-Z-DHA hybrid was synthesized following the procedure as previously reported [19]. In order to obtain the compound with suitable purity, the crude mixture was carefully chromatographed on a silica gel column in the presence of a pad of Florisil^®^ and the spectroscopic data are illustrated in Appendix A.

To compare the anticancer effect of UDCMe-DHA and DHA, HCC cells were treated with various concentrations of each compound for 72 h and cell viability was determined by the MTT assay. The dose–response curves of DHA (left panel) and UDCMe-DHA right panel) in HepG2 and Huh-7 cells are illustrated in Figure 2A. HepG2 cells were more sensitive to DHA and UDCMe-Z-DHA than Huh-7 cells. In HepG2 cells, the IC_50_ of DHA was 22.7 ± 0.39 μM (*n* = 3), while the IC_50_ of UDCMe-Z-DHA was 1.00 ± 0.20 μM (*n* = 3), which was ~23 times more potent than DHA (Table 1). In Huh-7 cells, the IC_50_ of DHA was 40.0 ± 1.34 μM (*n =* 3), while the IC_50_ of UDCMe-Z-DHA was 10.6 ± 2.28 μM (*n* = 3), which was ~4 times more potent than DHA (Table 1). Furthermore, in the same way as UDC-DHA [20], UDCMe-DHA was much less toxic to normal human dermal fibroblasts than DHA (Appendix A). Compared to DHA, UDCMe-Z-DHA was more potent against both HepG2 cells and Huh-7 cells, and showed a better selectivity toward cancer cells, therefore, the underlying mechanisms were further investigated.

### 2.2. UDCMe-Z-DHA Is More Stable Than DHA

We have reported that a BA-DHA hybrid UDC-DHA is more stable than DHA which may in part account for the increased anticancer activity of the hybrid [20]. To determine whether a similar scenario applied to UDCMe-Z-DHA, time course experiments were conducted to compare the cell viability of HepG2 cells treated with 20 μM DHA or 1 μM UDCMe-Z-DHA (the approximate IC_50_ of each after 72 h of treatment determined by the MTT assay) over time, and 1 μM DHA was also included for comparison. As shown in Figure 2B, only 20 μM DHA significantly decreased cell viability within 24 h, the effect was improved when the treatment was extended to 48 h, but 72 h treatment did not further increase growth inhibition. In contrast, the growth inhibitory effect of 1 μM UDCMe-Z-DHA was not apparent at 24 h, but significantly increased from 24 to 48 and 72 h, while 1 μM DHA did not exhibit any significant growth inhibitory effect even after 72 h. These results revealed that 1 μM UDCMe-Z-DHA acted gradually while 20 μM DHA exerted its effect more quickly to reach the same growth inhibitory effect at 72 h, suggesting that UDCMe-Z-DHA could be more stable than DHA.

HepG2 cells were next treated with 1 or 20 μM DHA, or 1 μM UDCMe-Z-DHA for 72 h, either with or without a daily change of medium with fresh compounds. The results indicated that a daily change of medium with 1 or 20 μM DHA more effectively suppressed cell growth compared to the no-change group; meanwhile, no differences were observed between the no-change and daily change groups treated with 1 μM UDCMe-Z-DHA (Figure 2C). Furthermore, when 20 μM DHA and 1 μM UDCMe-Z-DHA were preincubated in culture medium for 0–24 h at 37 °C before being added to HepG2 cells for 72 h, the activity of 20 μM DHA diminished quickly within 6 h in culture medium while the activity of 1 μM UDCMe-Z-DHA was not much affected even after 24 h of preincubation (Figure 2D). Chemical stability of UDCMe-Z-DHA in culture medium at 37 °C was assessed by HPLC-MS/MS analysis under comparable conditions previously used for DHA and UDC-DHA, which showed that UDC-DHA remained almost intact but DHA was degraded quickly and not detected at 24 h [20]. A time course of 0, 2, 15 and 24 h was conducted and UDCMe-Z-DHA was found ~95% intact up to 24 h in cell culture medium (Figure 2E). Taken together, these data suggested that in the same way as UDC-DHA [20], UDCMe-Z-DHA was more stable which may contribute to the increased potency.

### 2.3. The Effect of DHA and UDCMe-Z-DHA on Cell Cycle Progression and Apoptosis in HCC Cells

To compare the effect of DHA and UDCMe-Z-DHA on cell cycle progression, HepG2 and Huh-7 cells were treated for 24 or 48 h, followed by propidium iodide (PI) staining and flow cytometric analysis. The cell cycle distribution of HepG2 cells treated with 0.5–2 μM UDCMe-Z-DHA for 24 h is shown in Appendix A, and UDCMe-Z-DHA increased the G0/G1 population dose-dependently as highlighted in Figure 3A. At 48 h, both DHA (10–20 μM) and UDCMe-Z-DHA (0.5–2 μM) markedly induced subG1 populations in HepG2 cells in a dose-dependent manner (Figure 3B). Western blot analysis revealed that both 20 μM DHA and 1 μM UDCMe-Z-DHA significantly downregulated Rb and cyclin D, and upregulated hypophosphorylated Rb (Hypo Rb) after HepG2 cells were treated for 24 h compared with the vehicle control group, confirming that DHA and UDCMe-Z-DHA induced G0/G1 cell cycle arrest (Figure 3C). Western blot analysis of HepG2 cells treated with DHA and UDCMe-Z-DHA for 48 or 72 h showed that 1 or 2 μM UDCMe-Z-DHA induced the cleavage of PARP and caspase-3 more effectively than 20 μM DHA (Figure 3D), indicating that UDCMe-Z-DHA elicited apoptosis better than DHA.

In Huh-7 cells, 20 μM DHA, 10 and 20 μM UDCMe-Z-DHA significantly increased the G0/G1 population, and decreased S and G2/M cells after 24 h of treatment. At 48 h, 20–40 μM DHA and 10–20 μM UDCMe-Z-DHA all markedly induced subG1 populations, while 20 and 40 μM DHA also significantly decreased G0/G1 populations. Interestingly, 10 and 20 μM UDCMe-Z-DHA caused decreases in S and G2/M populations at both 24 and 48 h (Figure 3E). In HepG2 cells, 1–2 μM UDCMe-Z-DHA also slightly downregulated cells in S and G2/M phases after 24 h of treatment (Appendix A).

### 2.4. The Effect of DHA and UDCMe-Z-DHA on ROS Production in HepG2 Cells

Previous studies have demonstrated that DHA can induce ROS through the cleavage of the endoperoxide bridge [6]. We have reported that both DHA and UDC-DHA significantly induced ROS generation but with different magnitude and timing in HepG2 cells [20]. Here, we investigated the effect of DHA and UDCMe-Z-DHA on ROS production over time in HepG2 cells. As shown in Figure 4A, 40 μM DHA induced the highest level of ROS at 12 h with a geomean of 15.60 ± 0.62 (*n* = 3), but ROS diminished after 12 h and showed no difference relative to the control at 24 h. Interestingly, 2 μM UDCMe-Z-DHA induced ROS gradually and persistently with the highest level at 24 h (geomean: 14.5 ± 0.39, *n* = 3) (Figure 4A), and the level remained high even after 28 h of treatment (geomean: 13.4 ± 0.45, *n* = 3). Intriguingly, 2 μM UDCMe-Z-DHA induced ROS levels equivalent to those induced by 40 μM UDC-DHA and seemed to be more persistent since the geomean of ROS induced by 40 μM UDC-DHA was ~15 at 24 h, but declined to ~10 at 28 h [20].

An antioxidant N-acetylcysteine (NAC) was added to test whether the maximum ROS induced by DHA or UDCMe-Z-DHA was affected. HepG2 cells were treated with 40 μM DHA for 12 h or 2 μM UDCMe-Z-DHA for 24 h with or without 2 mM NAC. As illustrated in Figure 4B, although similar levels of ROS were induced by DHA and UDCMe-Z-DHA in the absence of NAC, interestingly, cotreatment with 2 mM NAC only reversed the ROS induced by DHA but not by UDCMe-Z-DHA. When 5 mM NAC was used, ROS generated by 2 μM UDCMe-Z-DHA was slightly reversed (data not shown). Similar results were found in the cell viability assay. Cotreatment with 2 mM NAC rescued HepG2 cells from growth inhibition caused by DHA and significantly reversed the cell viability by 18–20% (Figure 4C), whereas 2 mM NAC cotreatment only slightly affected growth inhibition caused by UDCMe-Z-DHA and reversed the cell viability merely by 2–4% (Figure 4D).

Taken together, these results demonstrated the involvement of ROS in the anticancer activity of DHA and UDCMe-Z-DHA in HepG2 cells. However, differences may exist between ROS induced by DHA and UDCMe-Z-DHA.

### 2.5. The Effect of DHA and UDCMe-Z-DHA on Depolarization of MMP in HCC Cells

It has been reported that ROS may affect mitochondrial membrane potential (MMP) [22]. The JC-1 assay was then conducted to evaluate depolarization of MMP induced by DHA and UDCMe-Z-DHA in HCC cells, and representative dot plots are shown in the upper panels and quantitative results are shown in the lower panels in Figure 5. As shown in Figure 5A, the percentage of cells with MMP loss (green fluorescence) was higher when HepG2 cells were treated with 20 μM DHA, 2 μM or 20 μM UDCMe-Z-DHA for 48 h than the vehicle control. In Huh-7 cells, the percentage of cells with MMP loss caused by 40 μM DHA, 10 μM or 20 μM UDCMe-Z-DHA was also significantly higher than the vehicle control as illustrated in Figure 5B. These results suggested that the dysfunction of mitochondria induced by DHA and UDCMe-Z-DHA may be a potential underlying mechanism of anticancer activity.

### 2.6. The Effect of DHA and UDCMe-Z-DHA on the MAPK Signaling Pathway in HepG2 Cells

The mitogen-activated protein kinase (MAPK) signaling pathway regulates diverse biological functions including cell proliferation, differentiation and apoptosis [23]. Furthermore, it has been reported that DHA may induce apoptosis via the activation of p38 MAPK in HL-60 leukemia cells [24]. We then examined whether the MAPK signaling pathway was related to the activity of DHA and UDCMe-Z-DHA. HepG2 cells were treated with 20 μM DHA, 1 and 2 μM UDCMe-Z-DHA for 48 h and 72 h, and then subjected to Western blot analysis. DHA and UDCMe-Z-DHA markedly upregulated phosphorylation of the extracellular signal-regulated kinase 1/2 (p-ERK1/2, T202/Y204) at 48 and 72 h, indicating the activation of ERK, and the effect of UDCMe-Z-DHA was more prominent than DHA (*p* < 0.05, p-ERK1/2 induced by 20 μM DHA vs. 1 or 2 μM UDCMe-Z-DHA for 48 h or 72 h). Phosphorylated p38 (p-p38, T180/Y182) was also clearly induced by UDCMe-Z-DHA (Figure 6A).

It has been reported that ERK1/2 plays a crucial role in a variety of biological processes including cell proliferation, apoptosis, survival and differentiation [25]. Whether upregulation of p-ERK1/2 by DHA and UDCMe-Z-DHA was pro-survival or pro-apoptotic was further clarified. As shown in Figure 6B, U0126, a MEK inhibitor, enhanced the growth inhibitory effect of DHA and UDCMe-Z-DHA in HepG2 cells, suggesting that the activation of the MAPK/ERK signaling pathway caused by DHA and UDCMe-Z-DHA may play a pro-survival function in HepG2 cells.

### 2.7. The Effect of DHA and UDCMe-Z-DHA on the AMPK/Autophagy Signaling Pathway in HepG2 Cells

When the cellular energy is consumed, AMP-activated protein kinase (AMPK) can be activated in response to conditions such as glucose starvation or mitochondrial dysfunction [26]. ROS generation increases MMP loss, and therefore affects the production of ATP, which may contribute to the AMPK activation via phosphorylation at Thr172. Western blot analysis revealed that 1 or 2 μM UDCMe-Z-DHA significantly increased p-AMPK at Thr172 in HepG2 cells after 48 or 72 h of treatment, while 20 μM DHA decreased p-AMPK at 48 h but slightly increased p-AMPK at 72 h (Figure 7A). It has been reported that activation of autophagy through AMPK signaling provides an important mechanism to attenuate environmental stresses such as oxidative injury [27]. LC3B-II, a marker for autophagy was significantly induced by 1 or 2 μM UDCMe-Z-DHA in HepG2 cells especially after 72 h of treatment. DHA exhibited a similar effect, but to a much lesser extent. Taken together, these data indicated that autophagy induced by UDCMe-Z-DHA may result from activation of AMPK (Figure 7A).

### 2.8. Autophagy Induced by UDCMe-Z-DHA May Be Associated with Apoptosis in HepG2 Cells

To evaluate whether autophagy triggered by DHA or UDCMe-Z-DHA led to cell protection or cell death, HepG2 cells were treated with 20 μM DHA or 2 μM UDCMe-Z-DHA with or without 10 μM chloroquine (CQ), an inhibitor of autophagy, for 72 h and then harvested for Western blot analysis. Apoptosis-related proteins including PARP and caspase-3 were examined to determine the correlation between autophagy and apoptosis. Treatment with CQ alone for 72 h significantly induced cleaved PARP and cleaved caspase-3. Compared with UDCMe-Z-DHA alone, the combination of UDCMe-Z-DHA and CQ significantly reduced both cleaved PARP and cleaved caspase-3, while the combination of DHA and CQ did not show any clear effect (Figure 7B). Thus, autophagy may play a protective role in HepG2 cells, but autophagy induced by UDCMe-Z-DHA may lead to apoptosis in HepG2 cells.

### 2.9. ROS Induced by UDCMe-Z-DHA May Cause Autophagy in HepG2 Cells

To determine the correlation between ROS and autophagy, HepG2 cells were treated with 20 μM DHA and 1 μM UDCMe-Z-DHA with or without 2 mM NAC for 72 h and harvested for Western blot analysis. As shown in Figure 6C, LC3B-II induced by DHA and UDCMe-Z-DHA was obviously reduced in the presence of 2 mM NAC, suggesting that ROS induced by UDCMe-Z-DHA may cause autophagy and eventually contribute to apoptosis (Figure 7B,C).

## 3. Discussion

DHA has the potential to be repurposed as an anticancer agent, but the application is limited owing to its short half-life. We reported previously that UDC-DHA was equally effective against both HepG2 and Huh-7 HCC cells with IC_50_ of 1.75 µM and 2.16 µM, respectively, which were 12.2 (HepG2) and 18.5 (Huh-7) times more potent than DHA [20]. In this study, we demonstrated that although UDCMe-Z-DHA (IC_50_ = 10.6 µM) was less effective than UDC-DHA, it was still ~four-fold more potent than DHA (IC_50_ = 40.0 µM) in p53-mutated Huh-7 cells. Significantly, UDCMe-Z-DHA (IC_50_ = 1.00 µM) was ~2-fold more active than UDC-DHA and ~23-fold more active than DHA (IC_50_ = 22.7 µM) in HepG2 cells (Figure 2A and Table 1). Furthermore, UDCMe-Z-DHA was much less toxic to normal fibroblast cells than DHA (Appendix A), suggesting that UDCMe-Z-DHA could be a safer drug candidate.

Artemisinins are chemically unstable and the antimalarial activity of DHA was almost completely eliminated after 24 h of incubation in plasma [28]. We have demonstrated that by conjugating DHA with UDCA via an ester linkage, the resulting UDC-DHA hybrid was much more potent than DHA in HCC cells which was in part due to increased stability relative to DHA [20]. We demonstrated here that similar to UDC-DHA, UDCMe-Z-DHA acted gradually and its activity was more stable in culture medium and possibly inside the cell as well (Figure 2B–E). The growth inhibitory effect of DHA was dramatic during the first 48 h, but no significant change was observed from 48 h to 72 h in the time course experiments. In contrast, UDCMe-Z-DHA exerted the cytotoxic effect gradually from 24 h to 72 h of treatment (Figure 2B). Furthermore, a daily change of culture medium with fresh DHA significantly increased its effect, but there was no significant difference between the no-change group and the daily change group of UDCMe-Z-DHA (Figure 2C). In addition, the activity of DHA diminished quickly after preincubation in culture medium for just 6 h, while the activity of UDCMe-Z-DHA was in a steady state after preincubation in the medium for up to 24 h (Figure 2D).

Based on our studies, UDC-DHA and UDCMe-Z-DHA displayed similar in vitro stability. In UDC-DHA, UDCA and DHA are conjugated by an ester moiety, while a triazole linkage is present in UDCMe-Z-DHA. Esterases are distributed in human liver, erythrocytes, plasma and the gastrointestinal tract, which may hydrolyze hybrids linked by ester bonds [29,30]. It has been reported that the triazole linker can enhance the biostability, bioavailability and activity of medicinal compounds. In addition to being resistant to enzymatic degradation, the triazole chain provides stability in vivo and has non-charged properties that may facilitate cell penetration and distribution in vivo [31,32]. Thus, UDCMe-Z-DHA could be a more stable and active hybrid in vivo. Further investigation is required to clarify whether UDCMe-Z-DHA and UDC-DHA would be more effective than DHA, and whether UDCMe-Z-DHA would be more stable and active than UDC-DHA in vivo.

DHA has been shown to induce G0/G1 arrest [10]. We found that UDCMe-Z-DHA increased the G0/G1 populations after 24 h of treatment in both HepG2 and Huh-7 cells (Figure 3A,E), which correlated with the Western blot data demonstrating significant downregulation of Rb and cyclin D1, and upregulation of Hypo Rb in HepG2 cells (Figure 3C). DHA has also been reported to induce G2/M arrest in HepG2 cells [33]. However, we observed that UDCMe-Z-DHA markedly decreased G2/M populations in Huh-7 cells (Figure 3E), and to a lesser extent in HepG2 cells (Appendix A).

UDCMe-Z-DHA significantly induced subG1 populations in HepG2 and Huh-7 cells treated for 48 h (Figure 3B,E). Although 1 μM UDCMe-Z-DHA induced less subG1 cells than 20 μM DHA, 2 μM UDCMe-Z-DHA showed a slightly better effect than DHA. Western blot analysis of HepG2 cells treated for 48 h and 72 h showed that DHA and UDCMe-Z-DHA significantly increased the cleavage of PARP and caspase-3, markers of apoptosis, and 1 or 2 μM UDCMe-Z-DHA was more effective than 20 μM DHA (Figure 3D). Taken together, these results demonstrated that both DHA and UDCMe-Z-DHA induced mainly G0/G1 cell cycle arrest, and apoptosis in HCC cells.

The proposed mechanism of the antimalarial action of DHA involves cleavage of the endoperoxide bridge, producing free radicals in the cells of the parasite and then alkylates and oxidizes proteins, resulting in the death of the parasite [6]. We found that 40 μM DHA and 2 μM UDCMe-Z-DHA increased cellular ROS levels in HepG2 cells over time but those induced by DHA peaked at 12 h while UDCMe-Z-DHA, in the same way as UDC-DHA [20], peaked at 24 h (Figure 4A). This may be correlated with the gradual and persistent action as well as increased stability of UDCMe-Z-DHA as illustrated in Figure 2B–E. Strikingly, 2 μM UDCMe-Z-DHA was sufficient to induce ROS as effectively and more persistently than 40 μM UDC-DHA [20]. Furthermore, although the maximum level of ROS generated by DHA was similar to UDCMe-Z-DHA, ROS induced by DHA but not UDCMe-Z-DHA were reversed by NAC in HepG2 cells (Figure 4B). Similarly, NAC reversed the growth inhibitory effect of DHA but not UDCMe-Z-DHA (Figure 4C,D), suggesting that ROS induced by UDCMe-Z-DHA may be somewhat different from those induced by DHA, possibly as a result of the hybridization, and could not be removed by NAC, a commonly used antioxidant but with limited ROS reactivity [34].

Excessive ROS accumulation may be harmful to cells, causing an imbalance of mitochondrial redox status, the opening of the mitochondrial permeability transition pore and the collapse in the mitochondrial membrane potential [22,35]. Compared to DHA, UDCMe-Z-DHA induced a higher degree of MMP loss than DHA either at a low (2 μM) or equal concentration (20 μM) in HepG2 cells (Figure 5A). UDCMe-Z-DHA also induced MMP loss in Huh-7 cells (Figure 5B). However, no ROS was induced by UDCMe-Z-DHA in Huh-7 cells (data not shown), and the reasons leading to mitochondrial dysfunction remain to be investigated.

Previous studies have shown that DHA induces autophagy in several types of cancer cells. The depolarization of MMP decreases the production of ATP, which may lead to the activation of AMPK [36,37]. In HepG2 cells, DHA and UDCMe-Z-DHA activated the phosphorylation of AMPK at Thr172 (Figure 7A), which may regulate ULK and Beclin involved in autophagy, providing an important mechanism of adaptation to environmental stresses. Autophagy is a catabolic process whereby autophagosomes form and fuse with lysosomes to degrade toxic or unwanted cytoplasmic components, recycling the degraded materials as a source of energy and nutrients in the anabolic pathway [38]. While the role of autophagy in tumor progression remains controversial, DHA-induced autophagy has been shown to possess different activity in various cancer cells, such as a protective function in HepG2/ADM cells [39], autophagic cell death in A549 cells [40] or promotion of apoptosis in multiple myeloma (MM) cells [41].

LC3 is a central protein in the autophagy pathway. LC3-I combines with phosphatidylethanolamine to form an LC3-phosphatidylethanolamine conjugate (LC3-II), which is involved in the formation of autophagosomes. Western blot analysis showed that DHA and UDCMe-Z-DHA upregulated LC3B-II after cells were treated for 72 h compared with the control group, indicating that both DHA and UDCMe-Z-DHA induced autophagy which was associated with AMPK activation (Figure 7A). Notably, UDCMe-Z-DHA induced much higher levels of LC3B-II than DHA. Western blot analysis revealed that the inhibition of autophagy by CQ significantly reduced both cleaved PARP and cleaved caspase-3 induced by UDCMe-Z-DHA. By contrast, CQ alone increased cleaved PARP and cleaved caspase-3 in HepG2 cells (Figure 7B), suggesting an interesting scenario that autophagy may play a protective role in HepG2 cells, but autophagy induced by UDCMe-Z-DHA may contribute to apoptosis of HepG2 cells as in MM cells [41]. The signaling pathways involved in the interplay between autophagy and apoptosis induced by UDCMe-Z-DHA in HepG2 cells remain elusive. As shown in Figure 6, UDCMe-Z-DHA significantly upregulated p-ERK, which may play a pro-survival role. UDCMe-Z-DHA also induced p-p38 but to a lesser extent. Further investigation is required to determine whether p-p38 is involved in apoptosis.

Previous studies have shown that ROS can act as signaling molecules to activate the initiation of autophagosome [42]. On the other hand, autophagy plays a role to reduce the oxidative damage caused by ROS through removing damaged proteins or organelles [43,44]. It has been reported that a BH-3 mimetic induces ROS-mediated autophagy in HCC cells [45]. Since both DHA and UDCMe-Z-DHA induced ROS (Figure 4A) and autophagy (Figure 7A), we examined whether ROS induced by DHA and UDCMe-Z-DHA caused autophagy in HepG2 cells by Western blot analysis. Interestingly, an ROS scavenger NAC significantly decreased LC3B-II induced by DHA or UDCMe-Z-DHA (Figure 7C), indicating that ROS induced by DHA and UDCMe-Z-DHA may lead to autophagy in HepG2 cells. However, as illustrated in Figure 4B, NAC did not reverse ROS induced by UDCMe-Z-DHA as determined by the DCFH-DA assay. It is likely that some ROS induced by UDCMe-Z-DHA may be different from those induced by DHA, and DCFH-DA may have limitations in ROS detection since several ROS may not oxidize DCFH-DA as significantly as H_2_O_2_ [46,47]. Alternatively, the effect of NAC may be independent of its anti-ROS activity [34]. Further investigation is required to clarify this issue.

## 4. Materials and Methods

### 4.1. Materials

Human HCC cell line HepG2 was obtained from the American Type Culture Collection (ATCC). Human HCC cell line Huh-7 was obtained from the Japanese Collection of Research Bioresources. Primary normal human dermal fibroblast (NHDF) cells (C-12302) and PromoCell Fibroblast Growth Medium (C-23020) were purchased from PromoCell (Heidelberg, Germany). Low-glucose Dulbecco’s Modified Eagle’s Medium (DMEM), high-glucose DMEM, fetal bovine serum (FBS), 2.5% trypsin, 200 mM L-glutamine and nonessential amino acids (NEAA), 3-(4,5-Dimethylthiazol-2-yl)-2,5-diphenyltetrazolium Bromide (MTT), JC-1 were purchased from Thermo Fisher Scientific (Waltham, MA, USA). Antibiotic–Antimycotic (containing 10,000 units/mL penicillin, 10,000 μg/mL streptomycin, and 25 μg/mL amphotericin B) was purchased from HyClone (Logan, UT, USA). DHA was purchased from TargetMol (Boston, MA, USA). Propidium iodide (PI) and 2′,7′-dichlorodihydrofluorescin diacetate (DCFH-DA) were obtained from Sigma-Aldrich (St. Louis, MO, USA) and RNase A was obtained from BioShop (Burlington, ON, Canada). N-acetylcysteine (NAC) was obtained from MedChemExpress (Princeton, NJ, USA). Hypo Rb and Rb antibodies were purchased from BD Biosciences (San Jose, CA, USA). Caspase-3, p-ERK1/2, ERK1/2, p-p38, and p38 antibodies were purchased from Cell Signaling Technology (Boston, MA, USA). PARP (H-250) and cyclin D1 (DCS-6) antibodies were purchased from Santa Cruz Biotechnology (Santa Cruz, CA, USA). Antibodies for p-AMPK, LC3B and GAPDH were purchased from GeneTex (Irvine, CA, USA). The γ-Tubulin antibody was purchased from Sigma-Aldrich (St. Louis, MO, USA). Secondary antibodies HRP-conjugated anti-mouse and anti-rabbit IgGs were obtained from Cell Signaling Technology (Boston, MA, USA).

### 4.2. Cell Lines and Cell Culture

HepG2 cells (ATCC HB-8065) were cultured in low-glucose DMEM supplemented with 10% FBS (*v*/*v*), 2 mM L-glutamine, and antibiotics including 100 units/mL of penicillin, 100 µg/mL of streptomycin and 0.25 µg/mL of amphotericin B. Huh-7 cells were cultured in high-glucose DMEM supplemented with 10% FBS (*v*/*v*), 2 mM L-glutamine, NEAA and antibiotics. Primary NHDF cells were cultured in PromoCell Fibroblast Growth Medium according to the manufacturer’s instructions.

### 4.3. Cell Viability Assay

Cells were seeded into 96-well plates (3–5 × 10^3^ cells/well), incubated overnight and then treated with 0–50 μM DHA or 0–10 μM UDCMe-Z-DHA for 24–72 h in HepG2 or Huh-7 cells, or treated with 0–100 μM DHA or UDCMe-Z-DHA for 72 h in NHDF cells. Cell viability was determined by the MTT assay and the absorbance was measured at 570 nm with 690 nm as a reference wavelength using the SpectraMax Paradigm Multi-Mode Microplate Detection Platform (Molecular Devices, San Jose, CA, USA). DMSO was used as the vehicle control for data normalization. IC_50_ values were calculated using GraphPad Prism software.

### 4.4. HPLC-MS/MS Analysis

The stability of UDCMe-Z-DHA was assessed by HPLC-MS/MS as previously reported [20]. A time course of 0, 2, 15 and 24 h was performed and the concentration at each time point was calculated relative to the initial solution at 0 h. The experiment was performed in triplicate. MS/MS (ESI+) parameters: UDCMe-Z-DHA precursor ion 776 [M + 23], product ion 730, cone voltage 30 eV.

### 4.5. Cell Cycle Analysis

Cell cycle distribution was determined by PI staining and flow cytometry. HepG2 cells and Huh-7 cells were seeded into 12-well plates (10 × 10^4^ cells/well), treated with indicated compounds for 24 or 48 h, harvested by trypsinization and fixed overnight in 70% (*v*/*v*) ethanol at -20°C. After centrifugation at 500× *g* for 5 min at 4 °C, cells were resuspended in 200 μL PI stain solution (25 μg/mL PI, 20 U/mL RNase A, 25 μM EDTA and 0.025% (*v*/*v*) Triton X-100) and incubated for 30 min on ice in the dark. The stained cells were then detected by FACSCalibur (BD Biosciences, San Jose, CA, USA) and the results were analyzed by FlowJo software (Tree Star Inc., Ashland, OR, USA).

### 4.6. Western Blot Analysis

Cells were seeded into 6-well plates (20 × 10^4^ cells/well) overnight, treated with indicated concentrations of compounds in culture medium for 24, 48 or 72 h, harvested by trypsinization and lysed in SDS sample buffer. Protein samples were separated by 10% or 12% SDS-PAGE, transferred onto PVDF membrane, and subjected to Western blot analysis. Signals were detected by enhanced chemiluminescence. Images were acquired and quantified using the ChemiDoc XRS system and Image Lab software (Bio-Rad Laboratories, Hercules, CA, USA).

### 4.7. DCFH-DA Assay

HepG2 cells were seeded into 12-well plates (10 × 10^4^ cells/well) overnight and then treated with 40 μM DHA or 2 μM UDCMe-Z-DHA in the absence or presence of 2 mM NAC in culture medium for various time periods. DCFH-DA at a final concentration of 10 μM was added to the cells 30 min before the termination of the incubation period at 37 °C. Cells were harvested by trypsinization, resuspended in cold PBS, and detected by FACSCalibur. The geomean of cellular ROS production was analyzed by FlowJo software.

### 4.8. JC-1 Assay

Cells were seeded into 12-well plates (20 × 10^4^ cells/well) overnight, treated with compounds for 48 h, and JC-1 dye at a final concentration of 5 μg/mL was added to the cells 30 min before the termination of the incubation period at 37 °C. Cells were harvested by trypsinization, resuspended in cold PBS and detected by FACSCalibur. The data were analyzed by FlowJo software.

### 4.9. Data Analysis

Data are presented as mean ± standard error of the mean (SEM) of at least three independent experiments. All data analyses were performed using Microsoft Office Excel 2019 software (Microsoft, Redmond, WA, USA) and GraphPad Prism 6 (GraphPad Software Inc., San Diego, CA, USA). The statistical significance of the data was evaluated by two-tailed Student’s *t*-test and *p*-values less than 0.05 were considered statistically significant.

## 5. Conclusions

HCC is one of the most common causes of cancer death in the world and urgently in need of novel and more effective treatment options. DHA is an antimalarial drug and has been reported to exert anticancer activities against various cancers but a short half-life limits further clinical applications. In this study, we found that UDCMe-Z-DHA, a hybrid of UDCMe and DHA via a triazole linkage, was ~23 times and ~4 times more potent than DHA in HepG2 cells and Huh-7 cells, respectively, in part due to increased stability. In addition, UDCMe-Z-DHA was much less toxic to normal cells. UDCMe-Z-DHA induced G0/G1 arrest, ROS, MMP loss and autophagy, which may subsequently lead to apoptosis. Thus, UDCMe-Z-DHA may be a potential drug candidate for HCC.

## Figures and Tables

**Figure 1 molecules-28-02358-f001:**
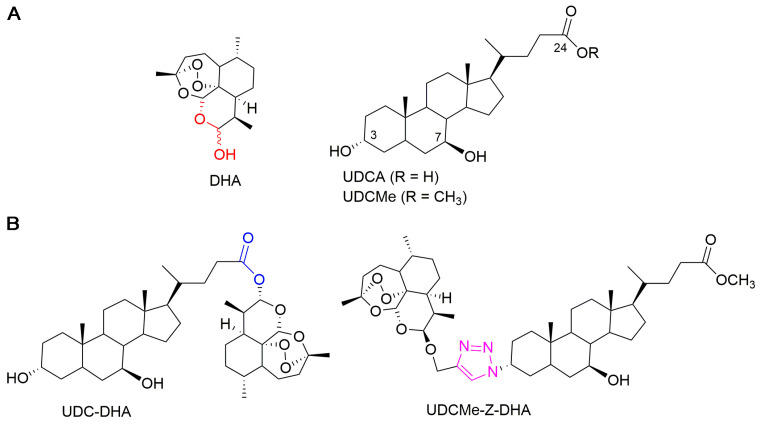
Chemical structures of parent molecules and BA-DHA hybrids. (**A**) Parent molecules DHA and UDCA. (**B**) UDC-DHA and UDCMe-Z-DHA hybrids.

**Figure 2 molecules-28-02358-f002:**
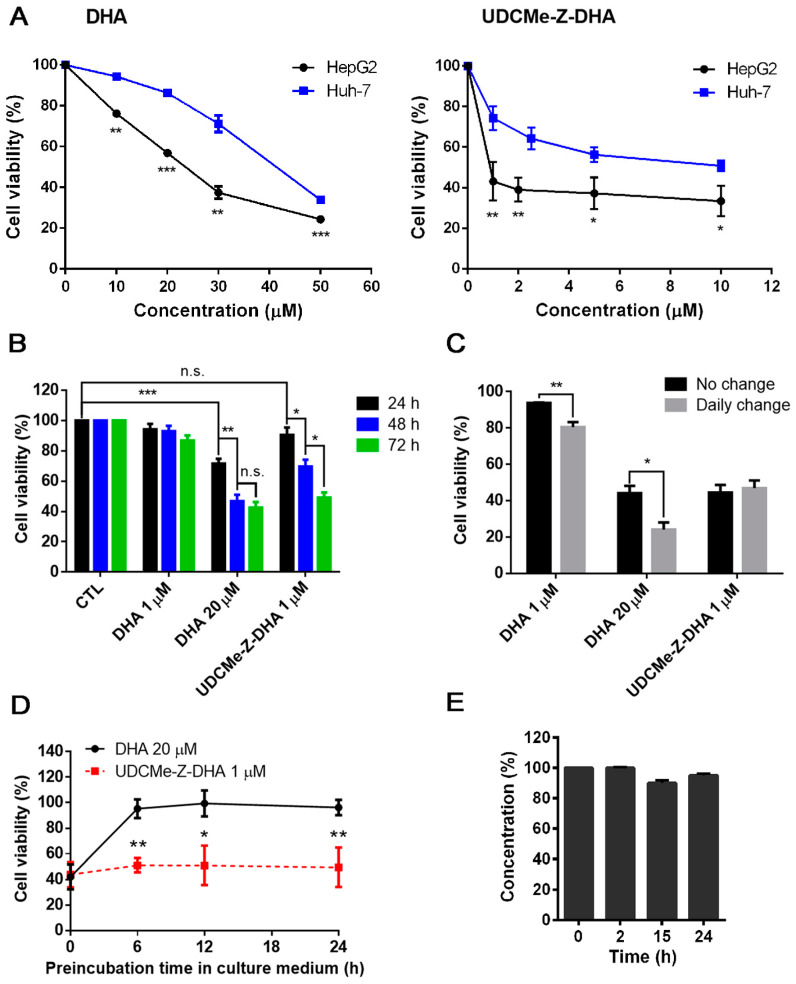
The growth inhibitory activity and stability of DHA and UDCMe-Z-DHA in HCC cells. (**A**) The effect of DHA and UDCMe-Z-DHA on the growth of HepG2 and Huh-7 HCC cells. Cells were treated for 72 h. HepG2 cells were significantly more sensitive to DHA and UDCMe-Z-DHA than Huh-7 cells. (**B**) Time-course study of DHA and UDCMe-Z-DHA in HepG2 cells. Cells were treated for 24–72 h. (**C**) The effect of daily change of DHA and UDCMe-Z-DHA on growth of HepG2 cells. Cells were treated for 72 h with no change or daily change of medium with fresh compounds. (**D**) Stability of DHA and UDCMe-Z-DHA in culture medium. Compounds were preincubated in medium for the indicated time periods at 37 °C before added to Hep2 cells for 72 h. The cell viability was measured by the MTT assay. (**E**) Chemical stability of UDCMe-Z-DHA in culture medium measured by HPLC-MS/MS. Data are presented as mean ± SEM of at least three independent experiments. Statistical significance was assessed by two-tailed Student’s *t*-test. * *p* < 0.05, ** *p* < 0.01, *** *p* < 0.001.

**Figure 3 molecules-28-02358-f003:**
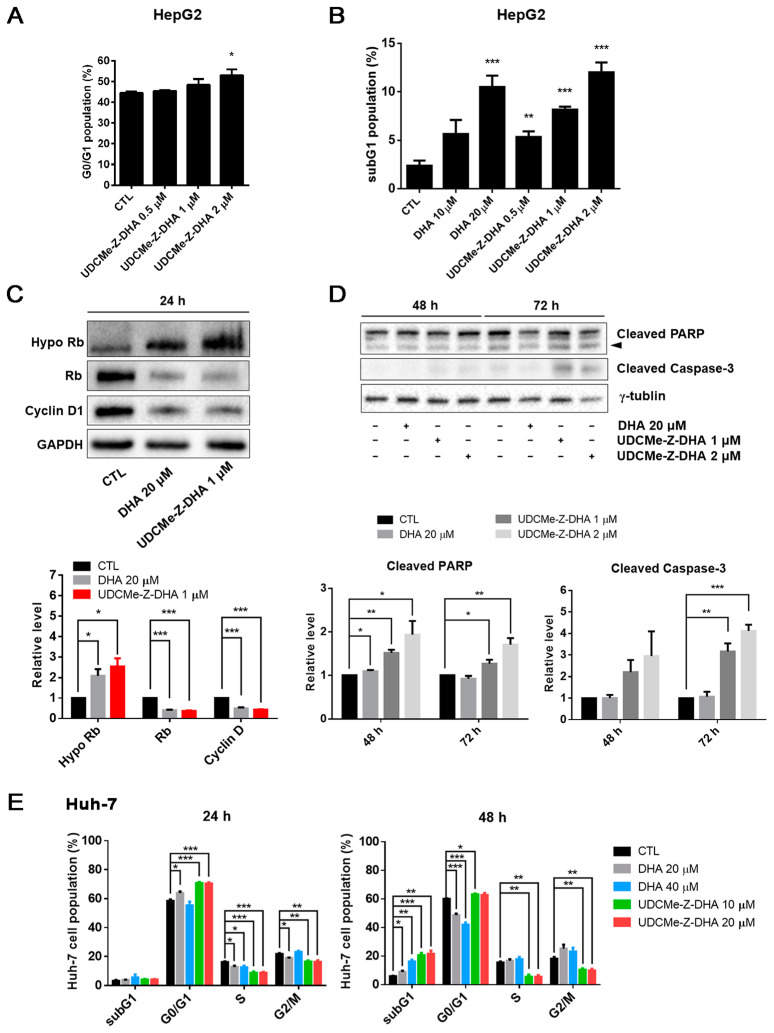
Effects of DHA and UDCMe-Z-DHA on cell cycle progression and apoptosis in HCC cells. (**A**) UDCMe-Z-DHA increased the G0/G1 population dose-dependently in HepG2 cells after 24 h treatment. Data are presented as mean ± SEM of 2–3 independent experiments. (**B**) DHA and UDCMe-Z-DHA significantly induced the subG1 population in HepG2 cells after 48 h treatment. (**C**) Rb and cyclin D1 protein levels in HepG2 cells treated with DHA and UDCMe-Z-DHA for 24 h. GAPDH was used as a loading control. (**D**) Apoptosis-related proteins in HepG2 cells treated with DHA and UDCMe-Z-DHA for 48 and 72 h. γ-Tubulin was used as a loading control. Representative images are shown in the upper panels and quantitative results are shown in the lower panels in (**C**,**D**). (**E**) Cell cycle distribution of Huh-7 cells treated with DHA and UDCMe-Z-DHA for 24 and 48 h. Data are presented as mean ± SEM of at least three independent experiments. Statistical significance versus the untreated vehicle control (CTL) was assessed by two-tailed Student’s *t*-test. * *p* < 0.05, ** *p* < 0.01, *** *p* < 0.001.

**Figure 4 molecules-28-02358-f004:**
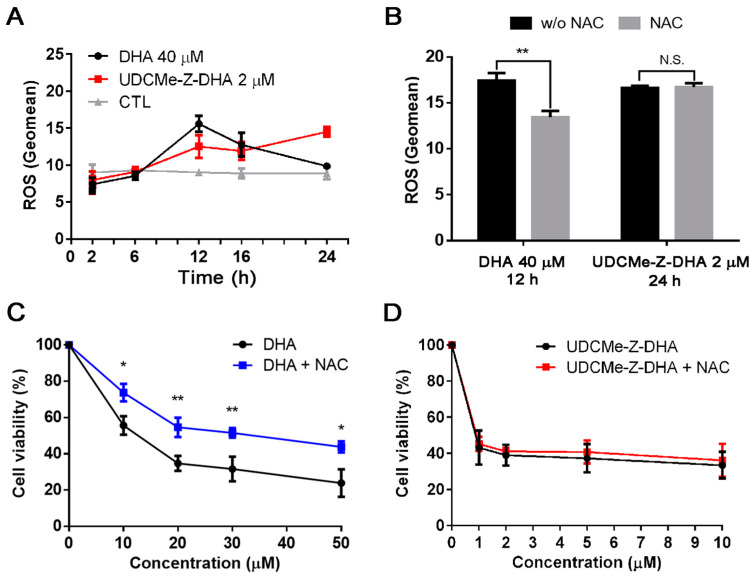
DHA and UDCMe-Z-DHA induce ROS in HepG2 cells. (**A**) Time course of ROS induction by DHA and UDCMe-Z-DHA. (**B**) NAC reversed ROS generation induced by DHA but not UDCMe-Z-DHA. (**C**) NAC reversed the growth inhibitory effect of DHA. (**D**) NAC did not affect the growth inhibitory effect of UDCMe-Z-DHA. In (**C**,**D**), HepG2 cells were treated with DHA or UDCMe-Z-DHA in the absence or presence of 2 mM NAC for 72 h, and cell viability was measured by the MTT assay. Data are presented as mean ± SEM of at least three independent experiments. Statistical significance was assessed by two-tailed Student’s *t*-test. * *p* < 0.05, ** *p* < 0.01.

**Figure 5 molecules-28-02358-f005:**
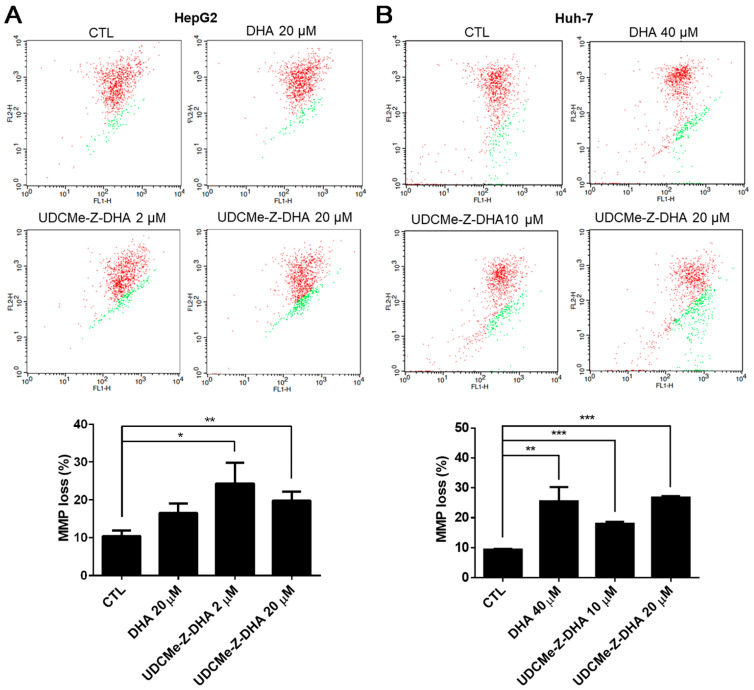
DHA and UDCMe-Z-DHA induced MMP loss in HCC cells. (**A**) HepG2 cells. (**B**) Huh-7 cells. HCC cells were exposed to indicated concentrations of DHA and UDCMe-Z-DHA for 48 h, and 5 μg/mL JC-1 was added to the cells 30 min before the termination of the incubation period at 37 °C. Cells were then harvested for flow cytometric analysis of JC-1 fluorescence. The percentages of cells with depolarization of MMP (labeled in green) were analyzed by CellQuest software. Data are presented as mean ± SEM of at least three independent experiments. Statistical significance was assessed by two-tailed Student’s *t*-test. * *p* < 0.05, ** *p* < 0.01, *** *p* < 0.001.

**Figure 6 molecules-28-02358-f006:**
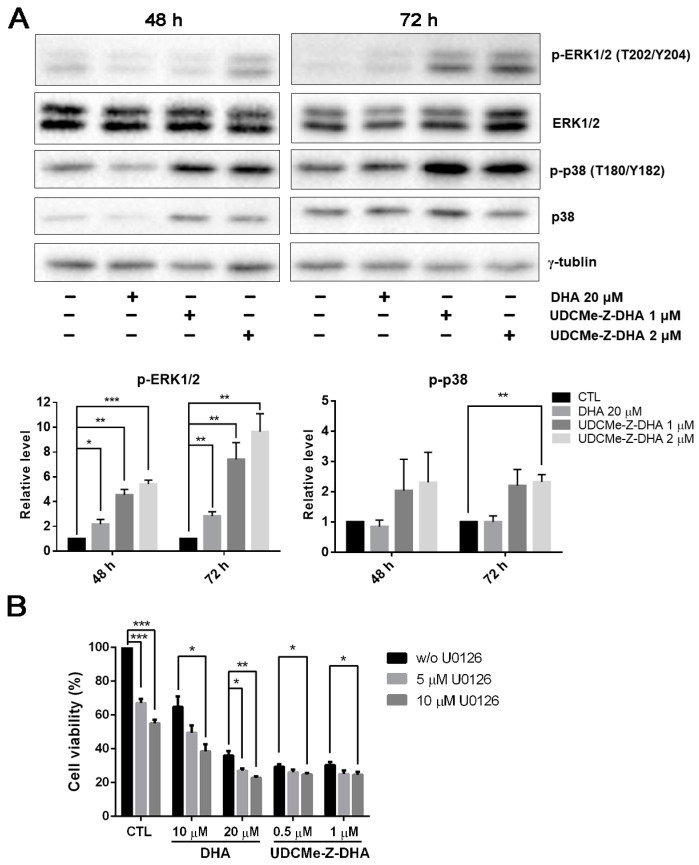
Effects of DHA and UDCMe-Z-DHA on the MAPK pathway in HepG2 cells. (**A**) The effect of DHA and UDCMe-Z-DHA on the MAPK pathway after 48 and 72 h of treatment. Representative images are shown in the upper panels and quantitative results of p-ERK1/2 and p-p38 are shown in the lower panels. γ-Tubulin was used as a loading control. (**B**) U0126 enhanced the growth inhibitory effect of DHA and UDCMe-Z-DHA. HepG2 cells were treated with DHA or UDCMe-Z-DHA in the absence or presence of U0126 for 72 h. Cell viability was measured by the MTT assay. Data are presented as mean ± SEM of at least three independent experiments. Statistical significance was assessed by two-tailed Student’s *t*-test. * *p* < 0.05, ** *p* < 0.01, *** *p* < 0.001.

**Figure 7 molecules-28-02358-f007:**
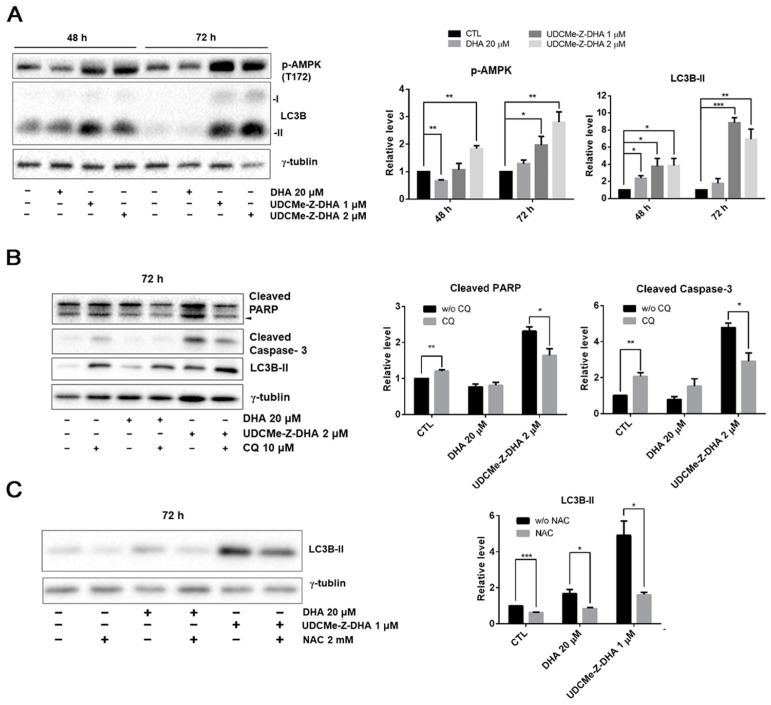
The effect of DHA and UDCMe-Z-DHA on autophagy and the interplay with ROS and apoptosis in HepG2 cells. (**A**) The effect of DHA and UDCMe-Z-DHA on the AMPK/autophagy signaling pathway in HepG2 cells. (**B**) Inhibition of UDCMe-Z-DHA-induced autophagy by chloroquine attenuated apoptosis. (**C**) Inhibition of ROS production by NAC attenuated autophagy induction by DHA and UDCMe-Z-DHA. Data are presented as mean ± SEM of at least three independent experiments. Statistical significance was assessed by two-tailed Student’s *t*-test. * *p* < 0.05, ** *p* < 0.01, *** *p* < 0.001.

**Table 1 molecules-28-02358-t001:** IC_50_ values and the ratios of IC_50_ values of DHA and UDCMe-Z-DHA in HepG2 and Huh-7 cells determined by the MTT assay.

Compound	HepG2		Huh-7	
	IC_50_ (μM) ^1^	Ratio ^2^	IC_50_ (μM) ^1^	Ratio ^2^
DHA	22.7 ± 0.39		40.0 ± 1.34	
UDCMe-Z-DHA	1.00 ± 0.20	22.7	10.6 ± 2.28	3.77

^1^ Cells were treated for 72 h and IC_50_ values were calculated based on cell viability measured by the MTT assay. Data are presented as mean ± SEM of three independent experiments. ^2^ The number in this column is the ratio of IC_50_ values of DHA and UDCMe-Z-DHA.

## Data Availability

Not applicable.

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
