# Peer review of "Anticancer Activity and Molecular Mechanisms of an Ursodeoxycholic Acid Methyl Ester-Dihydroartemisinin Hybrid via a Triazole Linkage in Hepatocellular Carcinoma Cells"

_molecules, 2023, doi:10.3390/molecules28052358_

Round 1

Reviewer 1 Report

The authors performed an interesting study in the field of synthesis of hybrid molecules based on bile acids and artemisinin. By the way, studies on the choice of bile acids as a structural element in the composition of a hybrid molecule, which facilitates the transport of the active substance in the tumor cell and allows easy penetration through the cell wall due to the high lipophilicity of the molecule, are quite common in the literature. In the introductory part, it would not be superfluous to give more examples of choosing such a strategy for the synthesis of hybrid molecules to justify the relevance and development of this approach in exploratory studies for the development of potential antitumor drugs. For example, even a cursory search of such literature revealed the following recent articles in this area - 1. B Brandes, S Hoenke, C Schultz, HP Deigner, R Csuk - Steroids, 2023 (https://doi.org/10.1016/j.steroids .2022.109148); 2.V.A. D'Yakonov, R.A. Tuktarova, L.U. Dzhemileva, S.R. Ishmukhametova, U.M. Dzhemilev, Pharmaceuticals, 14 (2) (2021), p. 84 (https://doi.org/10.3390/ph14020084); 3. Dang, Z.; Lin, A.; Ho, P.; Soroka, D.; Lee, K.-H.; Huang, L.; Chen, C.-H. Bioorg. Med. Chem. Lett. 2011, 21, 1926-1928; 4. Faustino, C.; Serafim, C.; Rijo, P.; Reis, C.P. Expert Opin. drug deliv. 2016, 13, 1133–1148 and others, which will not be superfluous to quote in this article.

In general, the article has a good chemical basis and reliable biological research, which allows me to give a positive conclusion about the possibility of its publication in Molecules. The article will be of interest to a wide range of researchers working in the field of natural compounds, pharmacology and medicinal chemistry.

Reviewer 2 Report

In materials and methods the cell viability assay is not mentioned the concentration of DHA and UDCMe-Z-DHA

DHA and UDCMe-Z-DHA induce ROS in HepG2 cells, in this experiment concentration are not mentioned in the methodology

Reviewer 3 Report

Review for molecules-2217711

In this research article entitled “Anticancer activity and molecular mechanisms of an ursodeoxycholic acid methyl ester-dihydroartemisinin hybrid via a triazole linkage in hepatocellular carcinoma cells”, the authors synthesized a series of bile acid-dihydroartemisinin hybrids and reported that UDC-DHA hybrid was 10-fold more potent than DHA against HepG2 hepatocellular carcinoma cells and showed that UDCMe-Z-DHA, a hybrid of ursodeoxycholic acid methyl ester and DHA via a triazole linkage, was more stable and more potent than UDC-DHA. These properties contributed to the anticancer effect for the assessed synthesized compound.

The study is well done and hereafter, just some minor comments revealed after reviewing the paper.

-       In lines 90-91, I think no need to indicate “manuscript in preparation” Authors can just delete that or insert “data not shown”, which is commonly used in some publications.

-       Part A of the Figure 2 should include statistical analyses other than the SEM. Some symbols such as asterisks should be inserted to exhibit the statistical differences.

-       Table 1, particularly in columns 3 and 5, the values do not report ratios.

-       What is “NAC” written in line 533? Author should indicate what NAC stands for?

-       What is the point of using both Excel and GraphPad Prism?

-       English language is fine but the manuscript should be revised to check for minor mistakes and punctuation, such as the additional “,” in lines 135, 136.

Reviewer 4 Report

I recommend major revision of this manuscript.

- My major concern is the similarity of the results with results previously obtained by the same authors for UDC-HDA. The advantages of UDCMe-Z-DHA over UDC-HDA need to be further highlighted in the discussion.

- Also, it is advised that the authors include experiments which compare between UDCMe-Z-DHA  and UDC-HDA.

- The enhanced stability and activity of UDCMe-Z-DHA  is observed in vitro and this has nothing to do with stability in vivo. Stability and activity need to be tested in an animal model, especially that the use of a hybrid drug is to enhance intestinal absorption, which is not present in this cell culture model.

- In figure 2, it is important to add UDC-DHA and compare UDCMe-Z-DHA to both DHA and UDC-DHA.

- In lines 115-119, indicate that these are the IC50 values at 74 h.

- Line 170, remove “like UDC” or add the reference or the experiments to show this.

- Figure 2E, perform the same HPLC analysis for HDA to show that it is degraded.

- Add 10uM and 20uM DHA to Figure 3A

- Discuss the statement in lines 244-247 in the discussion by mentioning why scavenging UDCMe-Z-DHA-generated ROS by NAC did not remove inhibition on cell proliferation and why NAC could not scavenge UDCMe-Z-DHA-generated ROS  “245 Taken together, these results demonstrated the involvement of ROS in the anticancer activity of DHA and UDCMe-Z-DHA in HepG2 cells. However, differences may exist between ROS induced by DHA and UDCMe-Z-DHA.

- Figure 6, do same experiment using an activator of ERK1/2 or p38.

Reviewer 5 Report

Please provide the results of all the parameters in the abstract

Please provide objectives of the study within abstract and last section of introduction 

In material and methods section authors used 96-well plates

Authors have to follow these comments 

Round 2

Reviewer 4 Report

My concerns have been addressed. The manuscript can be accepted after this minor change.

"Taken together, these data suggested that like UDC-DHA [20], UDCMe-Z-DHA was more stable which may contribute to the increased potency."

Modify to "Taken together, these data suggested that like UDC-DHA [20], UDCMe-Z-DHA was more stable than DHA which may contribute to its increased potency.